# Study on mechanical properties and permeability of elliptical porous scaffold based on the SLM manufactured medical Ti6Al4V

Chenglong Shi[1]*, Nana Lu[1], Yaru Qin[1]*, Mingdi Liu[1,2], Hongxia Li[1], Haichao Li[1]

**1** School of Chemistry and Chemical Engineering, Qinghai Nationalities University, Xining, China, **2** School of Chemical Engineering and Technology, Tianjin University, Tianjin, China

* qhislcas@126.com (CS); szdxqyr@126.com (YQ)

**Data Availability Statement:** All relevant data are within the paper.

**Funding:** This study was supported by Natural Science Foundation of Qinghai Province (No. 2020-

## Abstract

In this paper, we take the elliptical pore structure which is similar to the microstructure of cancellous bone as the research object, four groups of bone scaffolds were designed from the perspective of pore size, porosity and pore distribution. The size of the all scaffolds were uniformly designed as 10 × 10 × 12 mm. Four groups of model samples were prepared by selective laser melting (SLM) and Ti6Al4V materials. The statics performance of the scaffolds was comprehensively evaluated by mechanical compression simulation and mechanical compression test, the manufacturing error of the scaffold samples were evaluated by scanning electron microscope (SEM), and the permeability of the scaffolds were predicted and evaluated by simulation analysis of computational fluid dynamics (CFD). The results show that the different distribution of porosity, pore size and pores of the elliptical scaffold have a certain influence on the mechanical properties and permeability of the scaffold, and the reasonable size and angle distribution of the elliptical pore can match the mechanical properties and permeability of the elliptical pore scaffold with human cancellous bone, which has great potential for research and application in the field of artificial bone scaffold.

## Introduction

Natural bone is a kind of interconnected porous material, which is mainly composed of external dense cortical bone and internal soft cancellous bone [1]. The internal pore structure is very important for the infiltration, proliferation, differentiation and migration of cells [2, 3]. A large area of bone defect in human body needs external intervention [4]. Autografts and allografts are the common treatment of clinical bone defect diseases [5, 6]. However, due to the limited supply, easy to increase the risk of complications and other shortcomings, far from meeting the clinical needs [7, 8]. Therefore, it is very important to develop a suitable substitute for human bone tissue. All the studies mainly focus on two directions, one is the study of materials, the other is the study of structure, trying to match the original bone tissue through the

ZJ-394 953Q) and Natural Science Foundation of Qinghai Nationalities University (No. 2019XJG04).

**Competing interests:** The authors have declared that no competing interests exist.

combination of materials and structures, so as to facilitate the rehabilitation treatment of patients with bone defects.

Metal porous materials, ceramic porous materials, resin porous materials and other porous materials have become the key research objects during the recent years. Syahrom et al proposed body-centered cubic structure, face-centered cubic structure and tetrahedral plate structure [9]. Wang et al designed and prepared body-centered cubic structure of titanium alloy, and studied its mechanical properties [10]. Zhang et al studied diamond lattice pore structure [11]. Montazerian et al studied hexagonal and diamond dodecahedral structure of photopolymer resin. And the effect of pore shape on permeability was discussed [12]. Marques et al designed zirconia scaffolds with plate structure and studied their biological properties [13]. Ataee et al designed the commercial titanium gyroid structure and studied its mechanical compression response [14]. Alaboodi et al designed polycarbonate scaffolds with circular pore structure and studied their mechanical properties [15]. Kim et al studied the cellular response of bioglass composite scaffolds with rod structure [16]. L.Olivares et al proposed Gyroid and hexagonal scaffolds, and provides a computational approach to determine the mechanical stimuli at the cellular level when cells are cultured in a bioreactor and to relate mechanical stimuli with cell differentiation [17]. Zhao et al studied the effect of the flow velocity in the perfusion bioreactor on the mineralization of the cell mechanically stimulated by numerical simulation [18]. Although their research results showed that the designed bone scaffold can match some of the performance characteristics of human native bone, but the structure and morphology were quite different from the native bone tissue, which affects the migration, attachment, proliferation of bone cells inside the scaffold, and affects the subsequent bone tissue regeneration.

It is well known in the biomedical field that there are a large number of irregular pore structures in human bone tissue, and the enclosed arc-shaped pores have different curvatures at each point, and the shape is similar to an ellipse. Therefore, a more comprehensive study of bone scaffolds with elliptical pores is of great significance for exploring the idealized modeling of bone scaffolds.

Titanium alloy is widely used in surgical implants because of its good biocompatibility [19]. In this paper, Ti6Al4V is selected as the scaffold material. We focus on the mechanical properties and permeability of Ti6Al4V scaffolds.

In this study, based on the most basic uniform elliptical pore structure, four groups of representative bone scaffold structures with different pore size, porosity and different pore distribution characteristics were designed. The scaffolds were prepared one by one by selective laser melting (SLM) [20, 21]. The static compression properties and permeability of the four groups of scaffold samples were comprehensively evaluated by static compression experiment numerical simulation, hydrodynamic simulation and static compression experiment. The error of the forming process of SLM was discussed and analyzed by scanning electron microscope. The purpose of this paper is to make a comprehensive evaluation of the application of elliptical pore lattice structure in biomimetic bone scaffold and to evaluate the application potential of elliptical pore structure in biomimetic bone scaffold.

## Materials and methods

### Structure design method

The scaffold models were generated by UG 10.0 (Siemens PLM Software, USA). The design size of each model was $10 \times 10 \times 12$ mm, which was generated by Boolean cube elements with the same basic size, and the unit size was $1 \times 1 \times 1$ mm. The elliptical pores inside the elements were different, mainly reflected in the different radius of the major and minor axes and the different angles with the Y axis. These designs were mainly used to reflect the different pore size,

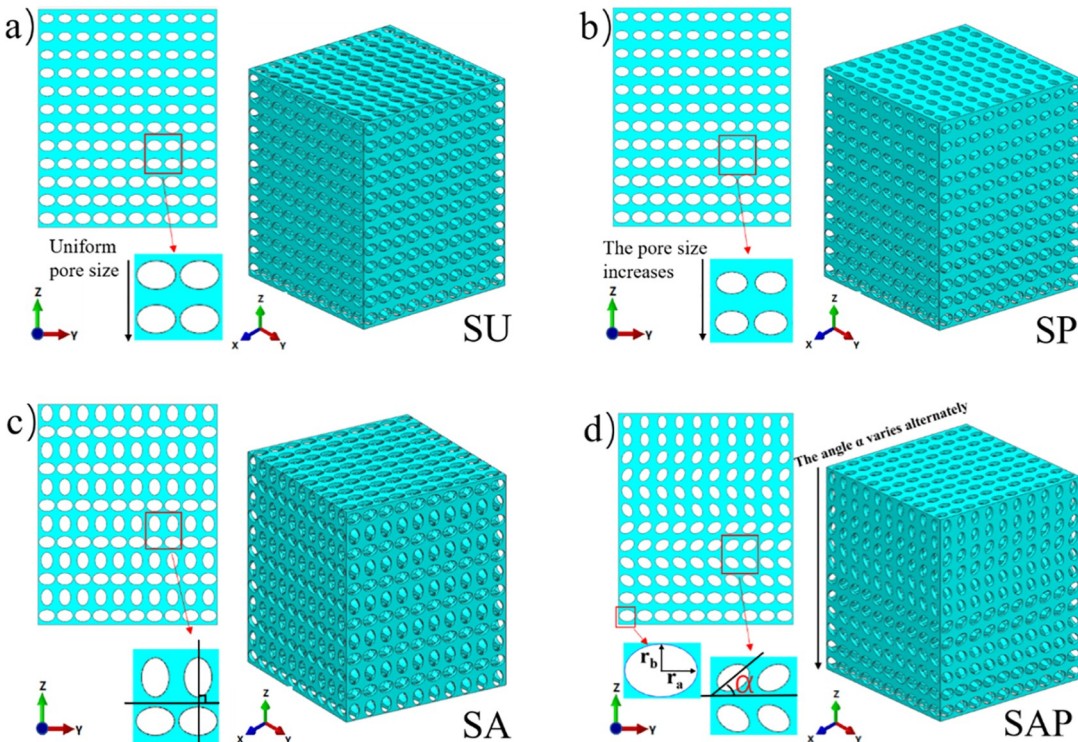

**Fig 1. CAD diagram of four groups of scaffolds.**

porosity and pore shape distribution of the scaffold. Fig 1 shows the CAD diagram of four sets of models. Uniform elliptical pore strut show in Fig 1A, the size of the elliptical hole is 900 μm for long axis and 650 μm for half axis, indicated as SU scaffold; Fig 1B show that, along the Z axis, the pore size of the strut decreases sequentially, the long axis and half axis decrease by 50 μm each time, the top two layers, the long axis is 650 μm, the short axis is 400 μm, indicated as SP scaffold; Fig 1C show that, elliptical pore size of the strut unchanged, adjacent two layers of elliptical pore orthogonal, indicated as SA scaffold. Fig 1D show that, the α angle and pore size of the strut change from top to bottom along the Z axis, which is same as that of Fig 1B. The design angle of the α angle changes from top to bottom, according to the order of 90˚, 60˚, -60˚, -30˚, -30˚, 0˚, indicated as SAP scaffold.

## Calculation method of porosity

Porosity is an important parameter in bone scaffold design, which is defined as the volume ratio of pore space in solid structure [22]. In this study, porosity can be expressed as the proportion of elliptical pore space in the unit cube. Using $V_E$ to represent elliptical pore volume and V to represent unit cube volume, then porosity (P) can be defined by Eq (1):

$$P = \frac{V_E}{V} \cdot 100\% \tag{1}$$

$V_E$ is the sum of the volumes of three elliptical cylinders, which can be calculated by multiple integrals. Considering the powerful measuring function of UG software, the volume of pore structure can be measured directly, and the theoretical porosity of the model sample can be obtained according to Eq (1).

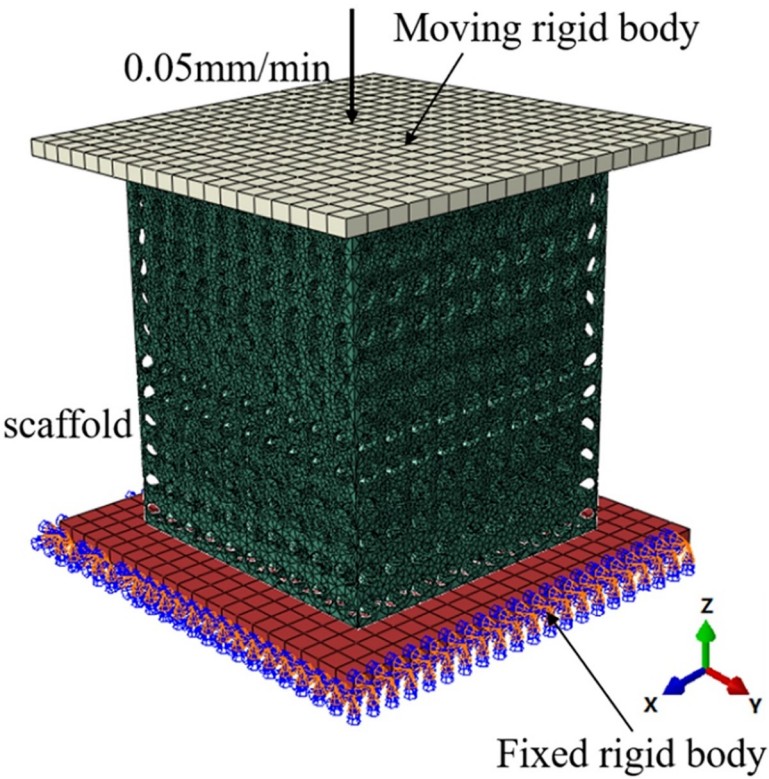

**Fig 2. Finite element analysis model of porous scaffold.**

The porosity of the samples after processing can be calculated by the density ratio. In this case, the porosity P can be calculated by Eq (2), where $\rho_o$ is the density of the solid alloy and $\rho$ is the density of the porous structure. The calculation method is the mass (m) of the structure divided by the volume (V).

$$P = \left(1 - \frac{\rho}{\rho_0}\right) \cdot 100\% \tag{2}$$

In this study, SU and SA scaffolds have the same porosity of 74%; The elliptical aperture of the SP and SAP scaffolds changes along the Z axis, so the porosity changes accordingly, from bottom to top as 74%, 68%, 62%, and 56%.

## Simulation analysis of mechanical compression test

As an important load-bearing structure, the compression property of implanted scaffold is important. Before the subsequent steps of the experiment, it is necessary to comprehensively evaluate the compression performance of the scaffold through numerical simulation. In this paper, the simulation process was completed in the software ABAQUS2016, and the boundary conditions were shown in Fig 2. The model was placed between the upper and lower steel plates, and the lower steel plate was a fixed steel plate, which imposes fixed constraints on the whole lower surface, and the upper steel plate was a movable steel plate, which plays the role of loading, and loads uniformly along the Z axis at the speed of 0.05 mm/min, which was consistent with the loading conditions of the micro-control electronic universal testing machine used in subsequent experiments. The scaffold was meshed by tetrahedral elements with the

**Table 1. Properties of Ti6Al4V materials in finite element analysis.**

| Materials | Density (kg/m$^3$) | Young's modulus (GPa) | Yield strength (MPa) | Poisson's ratio |
|---|---|---|---|---|
| Ti6Al4V | 4430 | 113.8 | 970 | 0.32 |

maximum mesh size of 0.05mm and the approximate global mesh size of 0.03mm. The upper and lower two steel plates were meshed by hexahedral elements. The whole process can be regarded as a quasi-static compression process. The loading conditions of the four groups of scaffolds were exactly the same, and the static characteristics of each group of scaffolds were calculated and compared. In the simulation process, the bone scaffold material is Ti6Al4V, material parameters as shown in Table 1 [23, 24].

## Preparation of model

Four groups of scaffold models were prepared by SLM process. Ti6Al4V powder was purchased from AVIC Maite Powder Metallurgical Technology (Beijing) Co., Ltd., with a particle size of 15 ~ 53 μm. 3D printing uses the YLMs-300 printing equipment of Jiangsu Yongnian Laser forming Technology Co., Ltd., which is equipped with a 500W laser with a spot size of 70 μm. The scanning thickness and scanning speed are 30 μm and 300 mm/s, respectively. The size of the model unit is 1 × 1 × 1 mm, and each sample is composed of 10 × 10 × 12 units. Four groups of representative models are shown in Fig 3.

## Characterization

In order to evaluate the manufacturing error of the SLM process and the processing effect of the scaffold samples. In this experiment, four groups of samples were scanned by SU3500

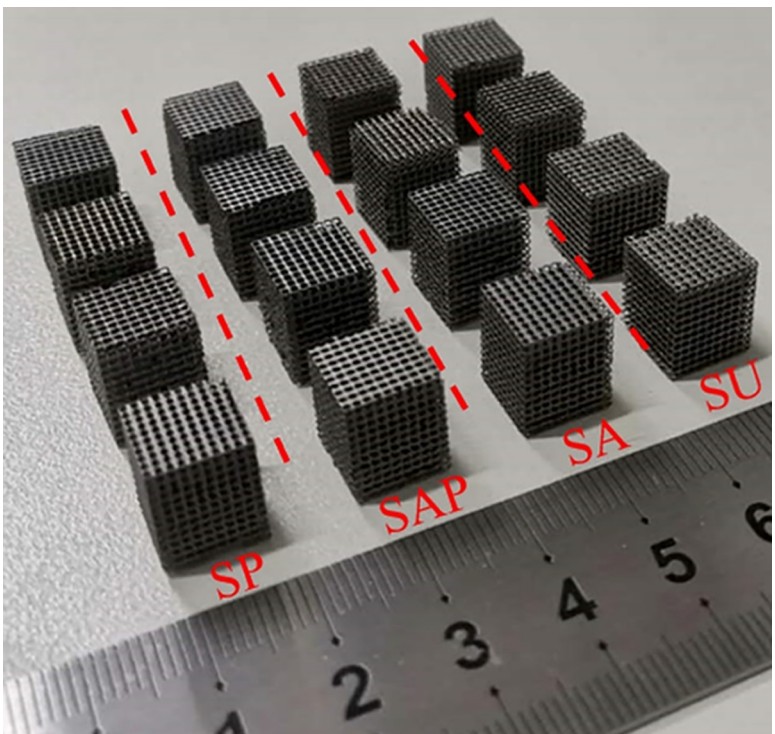

**Fig 3. Four groups of Ti6Al4V samples manufactured by SLM.**

(Hitachi, Japan) under the 15KV voltage using the scanning electron microscope SU3500. The image data of each group of samples were collected, and the elliptical pore size were measured. The measurement result was expressed as $D^*$. Selected multiple elliptical pores to measure and calculate the average value ($D_M$). Compared the measured average value with the design size (D) to evaluate the SLM manufacturing error. The deviation (η) can be calculated by Eq (3).

$$\eta = \left(\frac{D - D_M}{D}\right) \cdot 100\% \qquad (3)$$

## Mechanical property test

According to the International Standard for Mechanical Test, ductility Test, Compression Test of porous and porous Metals (ISO13314:2011), the compression tests of four groups of samples were carried out at the compression speed of 0.05 mm/min using a micro-controlled electronic universal testing machine (10 × 10 × 12 mm). Three specimens of each group were examined. The experimental process is shown in Fig 4. The force and deformation at each position were obtained. Through the analysis of the data, the stress-strain curves of each group of samples were drawn. The elastic modulus and yield strength of each group of samples were obtained to evaluate the mechanical properties of each group of samples.

## Fluid dynamics simulation analysis

Permeability is another important characteristic of bone scaffold, which can be measured by permeability [21]. Through the calculation of permeability, we can judge whether the designed structure matches the permeability of human cancellous bone. In this section, the permeability characteristic was studied by constructing a laminar computational fluid dynamics (CFD) model in Ansys Fluent solver (ANSYS Inc., PA, USA). The finite element volume method was used to solve the CFD model. The residual sensitivity criterion for convergence was set as 1e–5. Considering that the analysis object is an incompressible fluid with constant density, the Navier-Stokes equation defined by Eq (4) was used.

$$\begin{cases} \rho \dfrac{\partial \boldsymbol{v}}{\partial t} + \rho(\boldsymbol{v} \cdot \nabla)\boldsymbol{v} + \nabla P - \mu\nabla^2 \boldsymbol{v} = \boldsymbol{F} \\ \qquad\qquad \nabla \cdot \boldsymbol{v} = 0 \end{cases} \qquad (4)$$

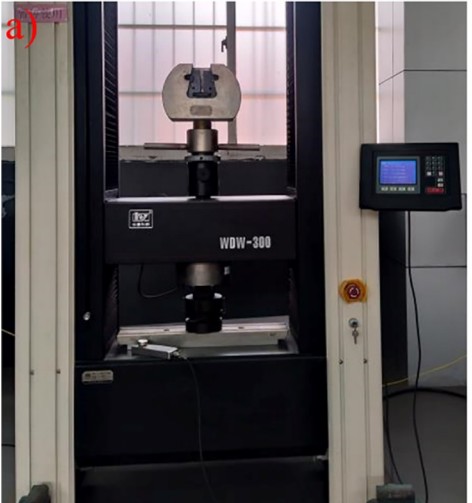
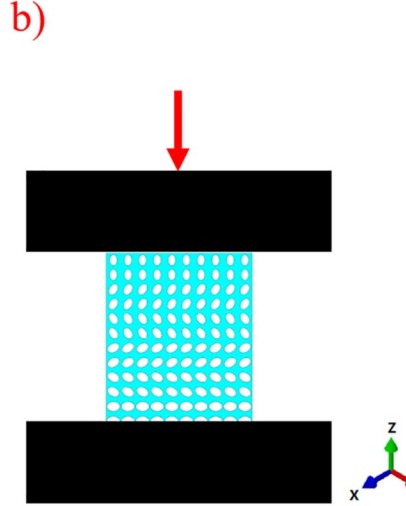

**Fig 4.** Mechanical compression test: (a) test operation diagram; (b) schematic diagram of test.

Where ρ is the density of the fluid (kg/m³); v is the velocity of the fluid (m/s); t is the time (s); ∇ is the operator; P is the pressure (Pa); μ is the dynamic viscosity coefficient of the fluid (Pa s); F is the acting force (N).

In order to simplify the simulation calculation and analysis, water was selected as the fluid domain material. At normal body temperature, the density and viscosity of water are 1000 kg/m³ and 1.45e-9 MPa.s respectively [25, 26]. The scaffolds were meshed using tetrahedral elements with the maximum mesh size of 0.001mm. The boundary condition of the fluid model was shown in Fig 5, the whole light color region is the fluid domain, and the green part was the scaffold model. Considering the influence of the boundary effect, a void area was added above the entrance of the CAD model, the height of the pore area was set at half of the model size height, and the inlet velocity applied to the scaffold was set to 1mm/s. Exit pressure is considered zero. The wall was assumed to have no slip [27].

Considering that in the four groups of scaffolds, SU and SA are symmetrical along the Z axis, while SP and SAP vary along the pore gradient along the Z axis, the fluid flow along the Z axis in different directions will show different flow characteristics. In this study, aiming at the two groups of asymmetric scaffolds of SU and SAP, in the fluid simulation, it was assumed that the fluid flows into the strut along the positive direction of the Z axis and the negative direction of the Z axis, respectively, and the two flow modes were compared.

Using ANSYS 18.2 to simulate the fluid flow in the bionic bone scaffold, the fluid flow law inside the structures were obtained, and the pressure difference between the inlet and outlet of the scaffolds and the permeability of the scaffold were calculated by Eqs (5) and (6).

$$\Delta P = P_{Plane\ A} - P_{Plane\ B} \tag{5}$$

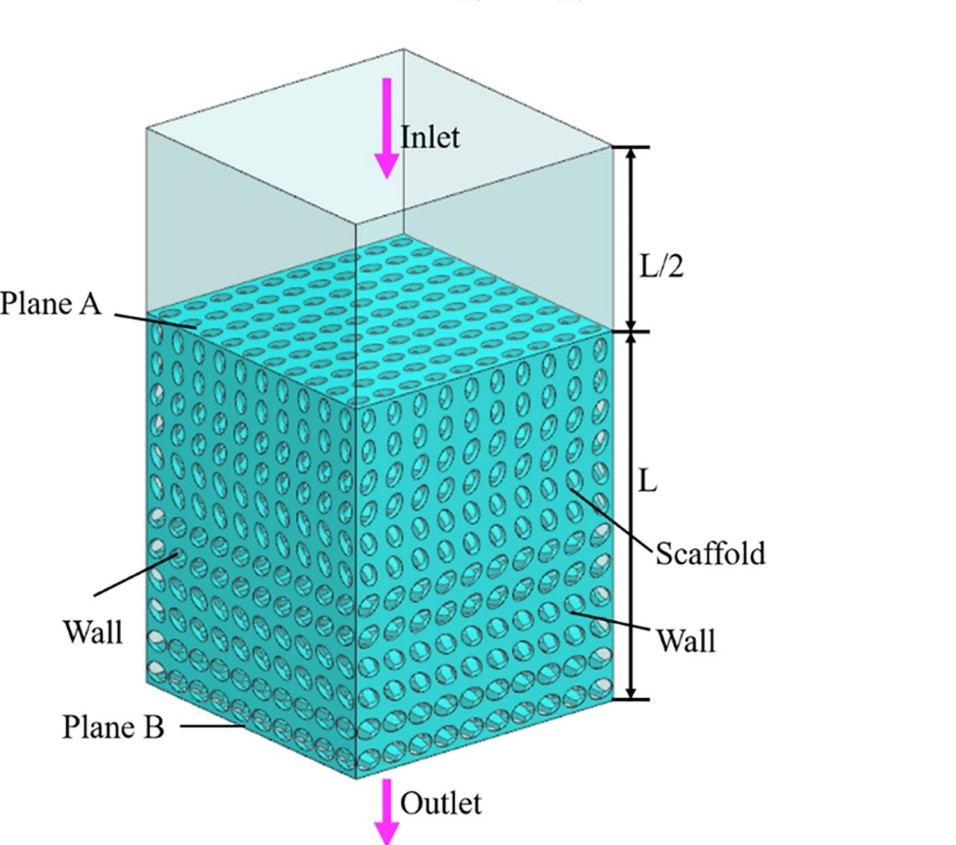

**Fig 5. Boundary conditions of CFD analysis.**

$$K = \frac{\mu \cdot v \cdot L}{\Delta P} \quad (6)$$

Where K is the permeability coefficient (mm2); L is the characteristic length (mm); $\Delta P$ is the pressure difference (MPa).

## Statistical analysis

Analysis was performed using SPSS 20.0 software (SPSS Inc., Chicago, Il, USA). All the data were expressed as mean ± standard deviation and analyzed with the one-way ANOVA. In all cases the results were considered statistically significant for $p < 0.05$.

## Results

### Characterization of the scaffolds

The structure of SAP sample is the most complicated, and theoretically it will cause greater manufacturing errors. Taking the SAP sample as an example, evaluate the manufacturing accuracy of samples. The scanning electron microscope (SEM) micrograph of the SAP samples manufactured by SLM is shown in Fig 6. It can be seen that the sample has certain manufacturing defects, but the overall structure is intact.

Three groups of elliptical apertures on the surface of the SAP scaffold were selected for measurement and error analysis was carried out. The results were shown in Table 2. The results show that there is a certain size error in the scaffold prepared by SLM, but the error was small, about 3%. The size of the elliptical aperture obtained by processing was slightly smaller than the design value.

### Simulation results of mechanical compression test

The finite element simulation results were shown in Fig 7. And Fig 7A reflect the stress distribution of the bone scaffold in the compression process, the stress distribution range was 25.36

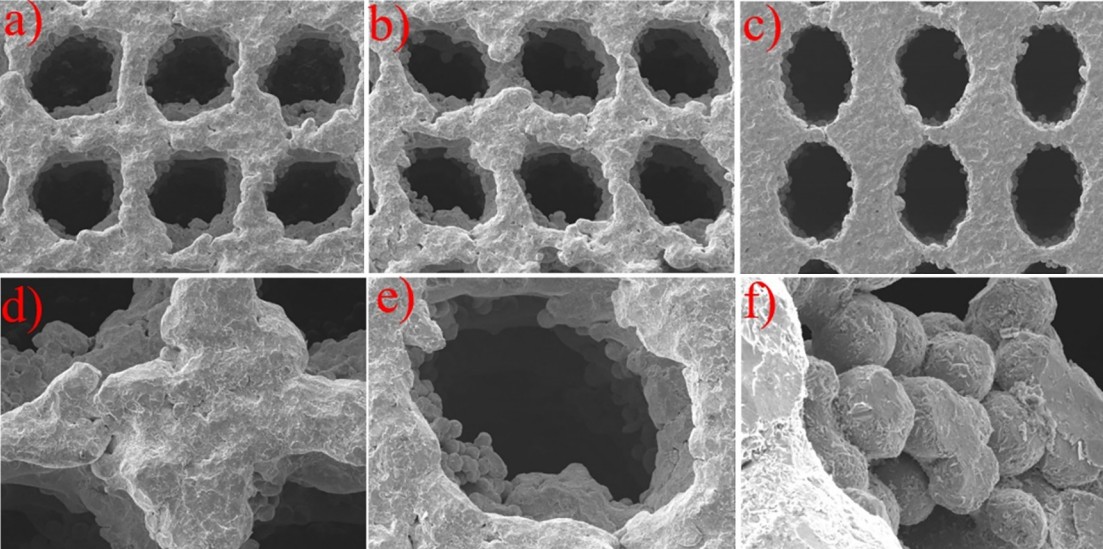

**Fig 6.** SEM morphology of SAP scaffolds: (a) and (b) side view of pores at different locations, (c) view of top surface, (d) micro-morphology of trabeculae, (e) single pore view, (f) view of the pore wall under a high-power microscope.

**Table 2. Measurement of elliptical pore size.**

| Elliptical pore size of SAP | Designed size D (um) | Measured size D* (um) | Mean size $D_M$ (um) | Deviation η (%) |
|---|---|---|---|---|
| major axis | 900 | 882 | 881.7 | 2.03 |
| | | 886 | | |
| | | 877 | | |
| | 800 | 785 | 783 | 2.13 |
| | | 781 | | |
| | | 783 | | |
| | 700 | 676 | 679 | 3.00 |
| | | 683 | | |
| | | 677 | | |
| minor axis | 650 | 626 | 633 | 2.62 |
| | | 631 | | |
| | 550 | 642533536 | 531.7 | 3.33 |
| | | 526 | | |
| | 450 | 438 | 436.3 | 3.04 |
| | | 435 | | |
| | | 436 | | |

to 7685 MPa, in which, most of the scaffold was in the lowest stress state, and the maximum stress was at the elliptical joint at the bottom of the scaffold. Fig 7B the displacement cloud diagram shows that under the compression of the external load, each layer of the bone scaffold has a similar compression state, and the crystal elements of the same layer had almost the same displacement change, indicating that the bone scaffold model has layer-by-layer compression deformation under the external load. The stress and strain of the four groups of bone scaffolds were simulated by finite element method as shown in the Fig 8A. It can be observed that the four groups of bone scaffolds had similar static behavior under external loading, in which, SAP scaffolds and SP scaffolds with lower porosity had higher stiffness and yield strength, SU scaffolds with larger porosity had the lowest stiffness and yield strength, while SA scaffolds with the same porosity as SU scaffolds had higher overall stiffness and yield strength than SU scaffolds. These results show that the overall stiffness and yield strength of SU scaffolds were higher than those of SU scaffolds. Both the porosity and the distribution angle of elliptical pores had influence on the overall stiffness and strength of the scaffolding. Smaller porosity and more diversified pore distribution can improve the static compression performance of the scaffold to a certain extent. The stress-strain curve obtained by finite element simulation was

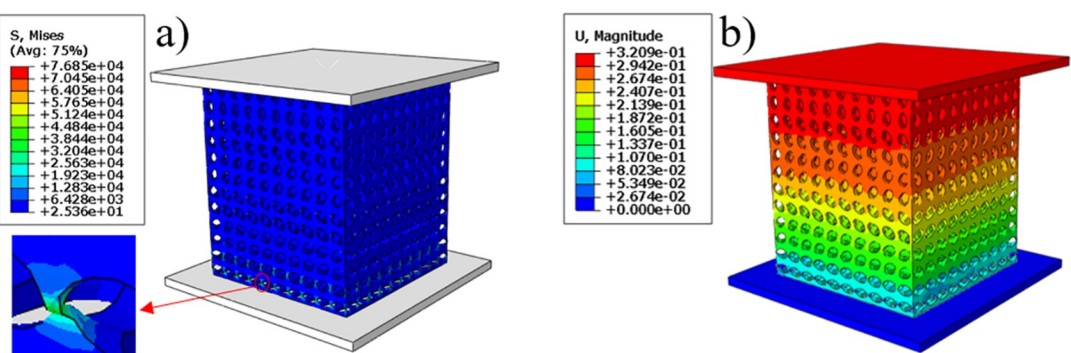

**Fig 7.** Finite element analysis cloud map: a) equivalent stress cloud map, b) displacement cloud map.

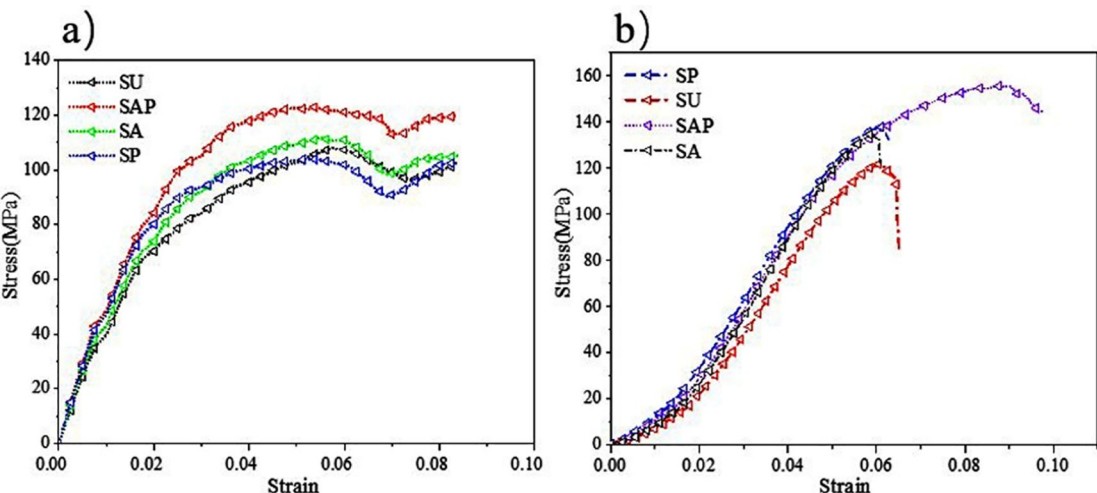

**Fig 8.** Stress-strain curve of scaffolds: a) Stress-strain curve by finite element simulation; b) Stress-strain curve obtained by compression test.

shown in Fig 8A, and the values of Young's modulus and yield strength were shown in Table 2.

## Results of mechanical compression test

The stress-strain curve of the scaffolds obtained by static compression test was shown in Fig 8B. It can be seen from the figure that the four groups of scaffold samples had similar static properties. The elastic modulus and yield strength of homogeneous SU scaffold were significantly lower than those of the other three groups, while SAP scaffold had higher elastic modulus and yield strength, which was largely similar to the results of finite element simulation. The mechanical properties of the support obtained by finite element simulation and mechanical tests were shown in Table 3. As can be seen from the results in the table, there was a certain difference between the results obtained by the two methods, which was a normal phenomenon. In addition, from the data in the table, it can be concluded that the porous structure can greatly reduce the elastic modulus of the scaffold, which was a feasible way to avoid stress shielding between the surgical scaffold and the bone.

## Permeability result analysis

The pressure difference between the inlet and outlet of each group of scaffolds and the movement characteristics of the fluid inside the scaffold can be obtained by the fluid simulation analysis, and the rationality of the structural design of the scaffolds can be evaluated. The pressure drop cloud picture of the four groups of scaffolds was shown in Fig 9. And Fig 9A was homogeneous SU scaffold pressure drop cloud. Fig 9B was SA scaffold pressure drop cloud

**Table 3. Static characteristics of four groups of scaffolds (FE = finite element).**

| | SU | SA | SP | SAP |
|---|---|---|---|---|
| Young's modulus by FE simulation (GPa) | 3.51 | 3.62 | 3.84 | 3.98 |
| Young's modulus by compressive testing (GPa) | 2.7±0.19 | 2.9±0.21 | 3.0±0.19 | 2.9±0.23 |
| Yield strength by FE simulation (MPa) | 103.36 | 107.48 | 104.25 | 121.62 |
| Yield strength by compressive testing (MPa) | 105±3.6 | 116±4.9 | 115±4.1 | 129±6.7 |

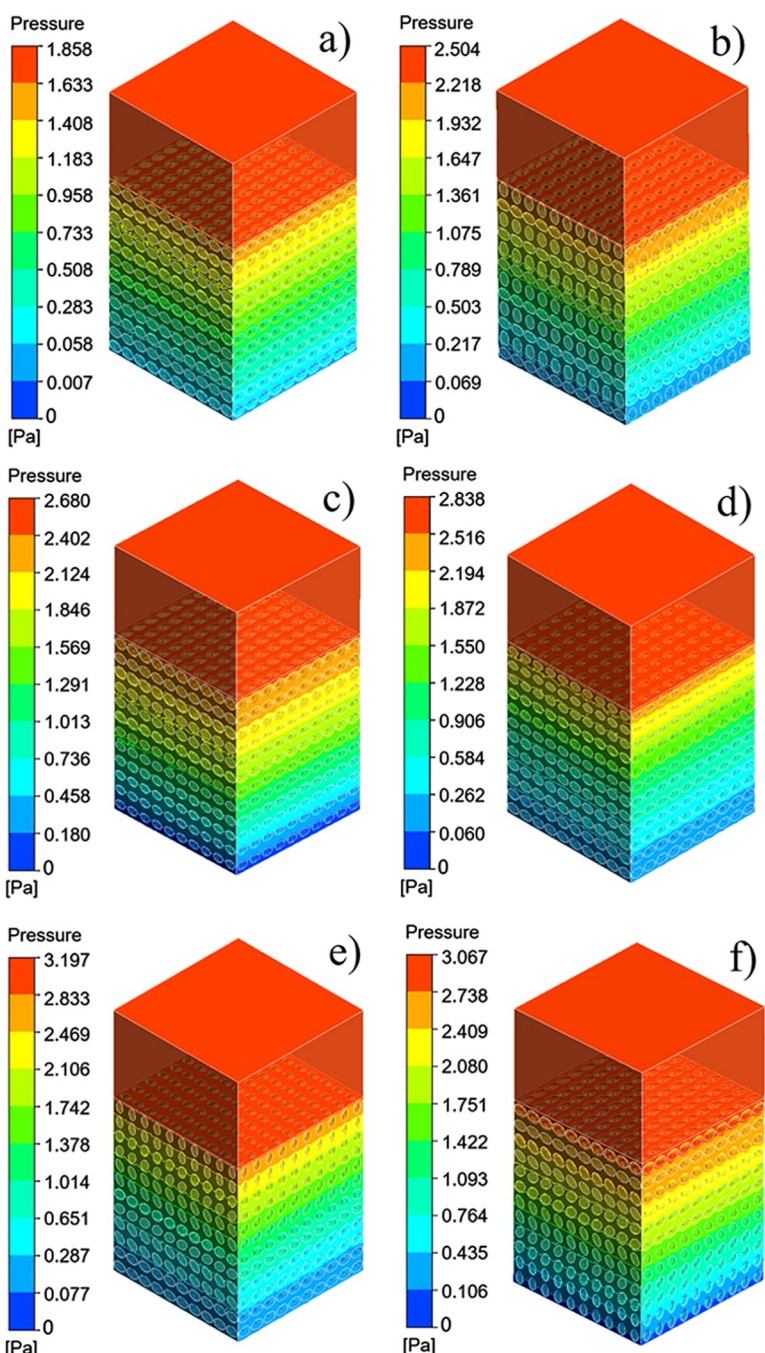

**Fig 9.** Four groups of scaffolds pressure drop cloud: (a) SU pressure drop cloud, (b) SA pressure drop cloud, (c) SP pressure drop cloud of direction +Z, (d) SP pressure drop cloud of direction -Z, (e) SAP pressure drop cloud of direction +Z, (f) SAP pressure drop cloud of direction–Z.

with orthogonal pores. Because the scaffolds of SP and SAP are asymmetrical along the Z axis, the flow characteristics of the fluid flowing into the scaffold along the Z axis are obviously different, so the flow characteristics of SP scaffolds and SAP scaffolds in +Z direction and -Z direction were studied respectively. Fig 9C and 9D were SP scaffold +Z direction and -Z direction pressure drop cloud images, Fig 9E and 9F were SAP scaffold +Z direction and -Z

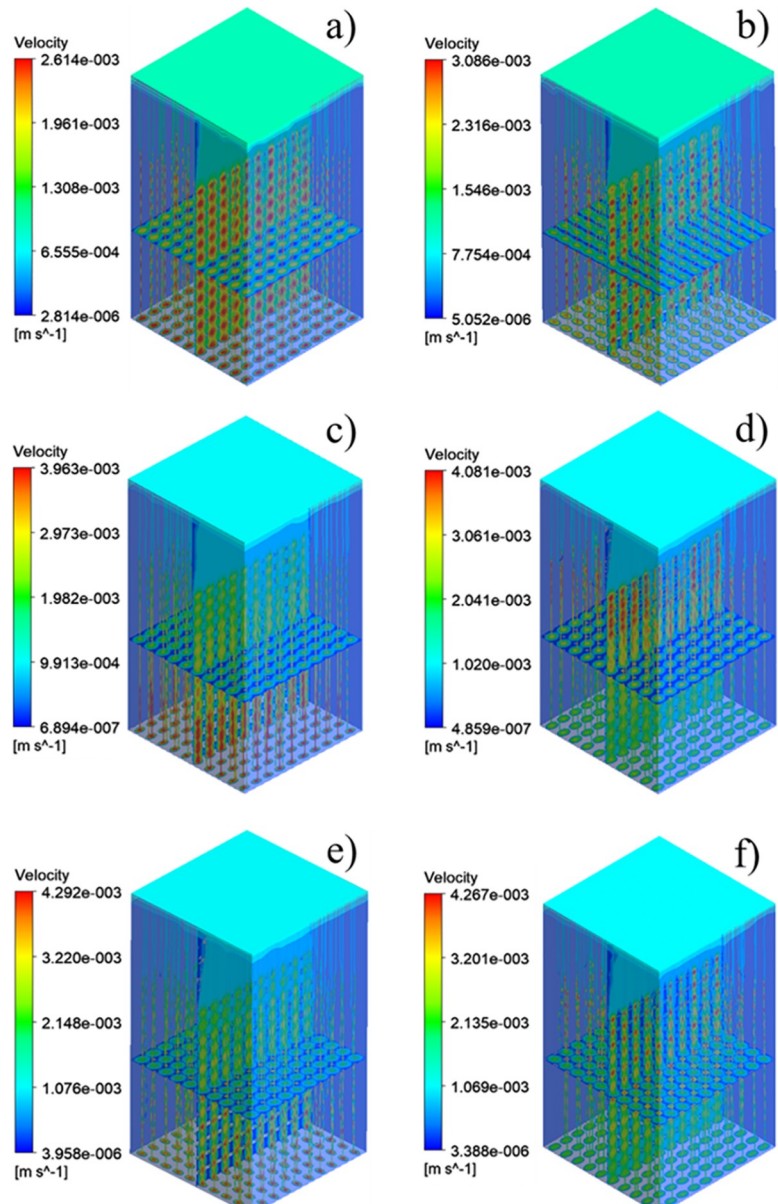

**Fig 10.** Four groups of scaffolds velocity cloud: (a) SU velocity cloud map, (b) SA velocity cloud map, (c) SP velocity cloud map of direction +Z, (d) SP velocity cloud map of direction -Z, (e) SAP velocity cloud map of direction +Z, (f) SAP velocity cloud map of direction–Z.

direction pressure drop cloud images respectively. Fig 10A–10F were the velocity cloud maps of the four groups of scaffolds and correspond to the pressure drop cloud images in Fig 9. In order to observe the flow characteristics inside the scaffold more intuitively, the velocity cloud images of the horizontal and longitudinal middle plane were intercepted in the velocity cloud images, and the internal flow charts were added. It can be seen from the figure that the flow velocity in the center of each group of scaffold unit was the highest, and the closer to the wall, the lower the velocity. For the homogeneous SU scaffold, each layer of the scaffold has the same flow characteristics, and the maximum flow velocity is the lowest in each group. The flow characteristics of each layer of SA scaffolds with orthogonal pores were similar, and the

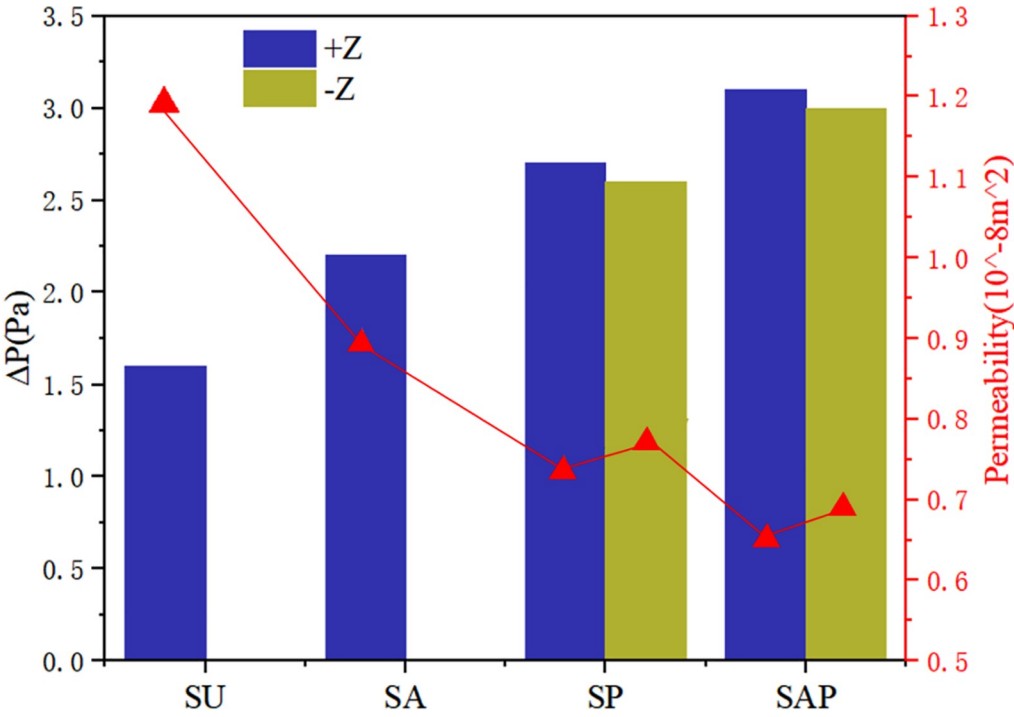

**Fig 11. Pressure drop and permeability of four groups of scaffolds.**

maximum flow velocity was larger than that of SU scaffolds. There was an obvious difference in the flow characteristics between the +Z direction and the -Z direction of the SP scaffold. Through Fig 10C, it can be observed that the internal velocity increases gradually along the +Z direction, reaches the exit position, and reaches the maximum. Through Fig 10D, it can be observed that the internal velocity was gradually decreasing along the -Z direction, and the velocity at the exit position was the lowest. Obviously, the flow along the +Z direction and -Z direction of the SAP scaffold was similar to that of the SP scaffold, but the flow of each layer of the SAP scaffold varies with the pore angle, and the maximum and minimum flow velocity were larger than those of the SP scaffold. From the analysis results of six groups of velocity cloud images, it can be seen that the gradient change of pore structure will bring about the gradient change of internal velocity, and the difference of pore angle also had a great influence on the velocity. In the gradient structure, taking the direction of smaller pores as the entrance and the direction of larger voids as the outlet, the fluid velocity at the entrance can be increased, which was beneficial to the migration of nutrients to the scaffold, and the velocity at the exit decreases. It was beneficial to the adsorption of nutrients in the scaffold, and the scaffold implantation was fixed in this way, which was likely to promote tissue repair and reconstruction.

Permeability quantifies the ability of a porous medium to transmit fluid through its interconnected pores or channels when subjected to pressure. The pressure drop and permeability calculation results of the six groups of flows were shown in Fig 11. From the results, the homogeneous SU scaffold has the minimum pressure drop and the maximum permeability, while the SAP scaffold with the change of pore and angle gradient has the maximum pressure drop and the minimum permeability. The permeability of SP scaffold and SAP scaffold along -Z direction was larger than that along +Z direction. Combined with the flow characteristics and permeability of the four groups of scaffolds, it can be seen that although the homogeneous

scaffold SU has a higher permeability, the fluid flow rate inside the scaffold is small, and the flow rate remains almost unchanged from the inlet to the outlet. While the design of the gradient structure, with smaller pores as the entrance direction (-Z), from the inlet to the outlet, the internal fluid flow rate gradually increases, which may accelerate the transportation of nutrients and the removal of waste. And the change of elliptical pore angle can change the internal velocity. These results show that the elliptical pore scaffold has high research and application potential as a surgical implant scaffold.

## Discussion

The purpose of present study was to investigate whether the bone scaffold structure with elliptical pores similar to the pores of human bone tissue can be used in the repair and reconstruction of human bone tissue. For this reason, four groups of bone scaffold structures with different porosity and different porosity distribution were designed. The static mechanical performance of the bone scaffold was evaluated mainly through statics simulation and experiment, and the permeability performance of the bone scaffold was evaluated by fluid dynamic simulation. This study was more comprehensive than the previous study by Syahrom and Wang et al on the cubic scaffold structure [9, 10]; Montazeriana et al on the hexagonal and rhombic dodecahedral scaffold structure [12]; and Alaboodi et al on the scaffold with circular aperture [15]. Their researches were more focused on the single compression performance or permeability performance of the scaffold structure, and the experimental results lack a direct comparison with the performance of human tissue. In the comparison of existing experimental results, the elastic modulus and yield strength of the gyroid bone scaffold designed by Ataee et al were 1465~2676 MPa and 44.9~56.5 MPa, respectively [14], which were close to the elastic modulus of trabecular bone. Zhang et al proposed a diamond cubic structure and designed multiple sets of Ti6Al4V bone scaffolds that can withstand a compressive strength of 36~140 MPa by changing the porosity. After being used in animal experiments, they have achieved good weight bearing effects [11]. However, they ignored the research on the permeability of the scaffold. The improvement of our research relative to them was that we have effectively evaluated the elastic modulus, yield strength, and permeability of the designed scaffold, and have made a comprehensive analysis of the obtained data results and the known performance parameters of human bone tissue. It was found that in terms of these three sets of indicators, the designed elliptical pore scaffold can meet the requirements. The calculation results of the elastic modulus and yield strength of the four sets of scaffolds are shown in Table 3. The elastic modulus of the SU, SP, SA and SAP scaffolds were 2510~2890 MPa, 2690~3110 MPa, 2810 ~3190 MPa and 2670~3130 MPa, respectively. The elastic modulus of human cancellous bone is in the range of 100~4500 MPa [28, 29], so it was within the elastic modulus of human cancellous bone; The elastic modulus of the SU, SP, SA and SAP scaffolds were 101.4~108.6 MPa, 111.1~120.9 MPa, 110.9~119.1 MPa and 122.3~135.7 MPa, respectively. The elastic yield strength of human cancellous bone ranges from 0.56~3.71 MPa [30, 31]. Obviously, the compressive strength of the four sets of scaffolds was better than that of cancellous bone. The static performance range of the structure designed by Ataee and Zhang et al was also close. The permeability of the four groups of scaffolds was shown in Fig 11. It was obvious that the permeability range was within the range of human cancellous bone permeability ($0.5 < k \, (10^{-8} \, \text{m}^2) < 5$) [32–34]. Of course, by comparing the static properties and permeability of the four groups of scaffolds, it was not difficult to find that porosity and pore distribution have a great influence on the static properties, permeability and fluid flow characteristics of the scaffold. We believe that in the SP and SAP structures with pore gradient changes, because the pore changes will affect the regional flow rate, the gradually increasing flow rate may accelerate the

supply of nutrients and the removal of waste. Of course, this may require more rigorous proof. These characteristics provide certain guidance for the subsequent further optimization of the scaffold and better biological performance. Of course, There were also some shortcomings in our research. The main shortcoming was the lack of research on the biological properties of the scaffold compared with the research of L.Olivares and Zhao et al [17, 18], which was our next research focus and direction.

## Conclusion

The main conclusions of this research are as follows:

1. SEM results show that the Ti6Al4V scaffold prepared by SLM has high accuracy. It is an effective method for manufacturing porous structures.

2. Mechanical simulation analysis and experimental results show that the four groups of elliptical pore scaffolds have the strength and rigidity that match the human native bone tissue, and SAP scaffolds have greater stiffness and yield strength.

3. Computational fluid dynamics ((CFD)) simulation results show that the four groups of scaffolds can meet the permeability requirements, but the four sets of scaffolds have different flow characteristics. It can be judged according to the flow characteristics that the SAP scaffolds with the change of porosity gradient and elliptical pore angle take the small pore as the entrance direction, which may accelerate the transportation of nutrients and the removal of waste.

These results show that reasonable design of porosity, pore size and pore distribution can make the performance of the elliptical pore scaffold achieve a higher match for the performance of human native bone tissue. There is great potential in the research and application of biomimetic bone scaffold, which is worthy of further exploration.

## Author Contributions

**Conceptualization:** Chenglong Shi.

**Formal analysis:** Nana Lu.

**Funding acquisition:** Chenglong Shi.

**Investigation:** Mingdi Liu, Hongxia Li, Haichao Li.

**Methodology:** Nana Lu, Mingdi Liu, Hongxia Li.

**Project administration:** Yaru Qin.

**Supervision:** Yaru Qin.

**Validation:** Yaru Qin, Mingdi Liu.

**Writing – original draft:** Chenglong Shi.

**Writing – review & editing:** Chenglong Shi, Nana Lu, Yaru Qin.

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
