## [Decision Letter · Decision Letter 0]

23 Dec 2020

PONE-D-20-36743

Study on Mechanical Properties and Permeability of Elliptical Porous Scaffold Based on the SLM Manufactured Medical Ti6Al4V

PLOS ONE

Dear Dr. Shi,

Thank you for submitting your manuscript to PLOS ONE. After careful consideration, we feel that it has merit but does not fully meet PLOS ONE’s publication criteria as it currently stands. Therefore, we invite you to submit a revised version of the manuscript that addresses the points raised during the review process.

We look forward to receiving your revised manuscript.

Kind regards,

Jose Manuel Garcia Aznar

Academic Editor

PLOS ONE

Journal Requirements:

'The authors are grateful for the financial aids from Natural Science Foundation of Qinghai Province (No. 2020-ZJ-953Q) and Natural Science Foundation of Qinghai Nationalities University (No. 2019XJG04).'

'Yes.

Shi CL received each award.

Natural Science Foundation of Qinghai Province (No. 2020-ZJ-953Q) website:http://kjt.qinghai.gov.cn/content/show/id/3063.

He is the first author of the article。'

Reviewers' comments:

Reviewer's Responses to Questions

**Comments to the Author**

1. Is the manuscript technically sound, and do the data support the conclusions?

Reviewer #1: Yes

Reviewer #2: Partly

2. Has the statistical analysis been performed appropriately and rigorously? 

Reviewer #1: No

Reviewer #2: Yes

3. Have the authors made all data underlying the findings in their manuscript fully available?

Reviewer #1: No

Reviewer #2: Yes

4. Is the manuscript presented in an intelligible fashion and written in standard English?

Reviewer #1: Yes

Reviewer #2: Yes

5. Review Comments to the Author

Reviewer #1: Dear authors,

Here my comments and suggest.

The authors make it clear from the beginning that we will meet in the study. Already from the title "Study on Mechanical Properties and Permeability of Elliptical Porous Scaffold" for a specific manufacturing process "SLM" and specific material "Ti6Al4Vand".

The study has been correctly planned, for this reason I propouse a minor revision for the study.

But I would like to know from the author:

-What is the statistical approach carried out for mechanical test?

-How calculate the standard deviation from the finite element models?

-What is the cell behavior in the material suface for the new roughtness created after fabrication?

-Do you know the integration of this porous materials and bone. Have some background?

-Why do you use Von misess stress and not the major principal stress.

-How is the differences found with bone tissue in term of permeability and mechanical properties. Do you know the zone in the body where do you can implant the scafolds?

Another minor recommendations are:

-In the introduction and discussion I recommend introduce more studies based on tissue engineering. Similar to theses examples:

https://www.sciencedirect.com/science/article/abs/pii/S0142961209007777

https://pubmed.ncbi.nlm.nih.gov/16003400/

-The color scale values must be corrected; removing (e + 000) and correcting the font size.

Thanks and Best wishes

Reviewer #2: In this paper, the authors proposed a porous scaffold with elliptical pore shape, which is considered as similar to cancellous bone in terms of permeability and mechanical strength. A combination of computational and experimental approaches have been used by the authors for characterising the permeability and mechanical compressive strength. This is a thorough and well-designed study. The outcome from this study could be useful for the R&D of bone implants. However, the following concerns / comments need to be properly addressed before accepting for publication.

1.Line 10 – 11: please specify compact bone or cancellous bone.

2.Line 36 – 45: I would suggest to summarise those studies in a table.

3.Line 47 – 49: What is the specific limitations of these previous studies? Why the bone cell behaviours will be different within the scaffold structures that are different from native bone?

4.In the introduction section, I don't think Fig. 1 is necessary, as it is commonly known in the biomedical field.

5.Line 58 – 59: Please justify why you chose to investigate these 2 parameters?

6.Line 65: “static compression experiment” is repetitive.

7.In section 2.3 – Simulation analysis of mechanical compression test: the mesh information of the FE model is missing.

8.In section 2.7 – Fluid dynamic simulation analysis: the flow properties, mesh and solver information are missing in this section, for example,

(1) Flow property: is the flow laminar or turbulent? What is the Re number?

(2) Mesh: what is the mesh strategy that is used for this CFD model? Why did you choose that mesh strategy? Have you done any mesh sensitivity analysis? If yes, what are the results?

(3) Solver: which numerical method is used, finite element method / finite volume method / finite difference method? What is convergence criteria of the CFD model?

Without clarifying these, it is hard to assess the accuracy of your CFD results.

9.In equation (4), please distinguish the vector, tensor, scalar in the equation. In Navier-Stokes equation, the expression of divergence of velocity seems wrong.

10.In Fig. 6 – BCs of CFD analysis: why would you define the inlet with L/2 away from the Plane A, and outlet at the Plane B?

11.In Fig. 9 and 10: in mechanical compression (both numerical and experimental), please justify why you chose the strain range of 0% - 9%? Are you sure bone can withstand such high strain, if you want to mimic the physiological level?

12.I suggest to combine Fig. 10 and fig. 9 together into one figure, which will more clearly show the dis/agreement between numerical and experimental results.

13.Line 295 “transmission performance”: Do you mean nutrient diffusion? In nutrient diffusion, permeability will influence the diffusivity. But permeability itself can not be used for directly characterising the diffusion. Please revise this sentence.

14.Line 299 – 301: should this be in discussion section?

15.Line 304 – 307: How can you reach this point of migration of nutrient? Which parameter is it based on? If you are based on fluid velocity, I only can see that the local fluid velocity is changed with the change of local pore geometric features (pore size, porosity, pore shape), but can not see how this can influence the overall nutrient delivery throughout the scaffold.

16.After the results section, I would suggest to add a section of Discussion, in which the obtained results need to be compared to the references. Otherwise the current version of conclusion is not convincing.

17.Line 336: What does “beneficial to the migration and adsorption of nutrients to the scaffold” mean? Do you mean enhanced nutrient diffusion? Please clearly specify it.

18.Line 337 – 340: Without comparing with other already available scaffold geometries, how can you get this conclusion?

6. PLOS authors have the option to publish the peer review history of their article (what does this mean?). If published, this will include your full peer review and any attached files.

Reviewer #1: **Yes: **Andy L. Olivares

Reviewer #2: **Yes: **Feihu Zhao

---

## [Author Response · Author response to Decision Letter 0]

6 Jan 2021

Dear Editors and Reviewers:

Thank you for the reviewers’ comments concerning our manuscript entitled “Study on Mechanical Properties and Permeability of Elliptical Porous Scaffold Based on the SLM Manufactured Medical Ti6Al4V” (ID: PONE-D-20-36743). Those comments are all valuable and very helpful for revising and improving our paper, as well as the important guiding significance to our researches. We have studied the comments carefully and have made correction which we hope meet with approval. Revised portion are marked in blue in the paper. The main corrections in the paper and the responds to the reviewer’s comments are as flowing: 

Responds to the reviewer’s comments:

Reviewer #1: Dear authors,

 Here my comments and suggest

The authors make it clear from the beginning that we will meet in the study. Already from the title "Study on Mechanical Properties and Permeability of Elliptical Porous Scaffold" for a specific manufacturing process "SLM" and specific material "Ti6Al4Vand".

The study has been correctly planned, for this reason I propouse a minor revision for the study.

But I would like to know from the author:

(1) What is the statistical approach carried out for mechanical test?

Response: Sorry, we forgot to add a comment on the statistics approach, All the data were with the one-way ANOVA. The specific content has been added in the section 2.8. (Page 11, Lines 199-202)

(2) How calculate the standard deviation from the finite element models?

Response: We did not calculate the standard deviation of the finite element model, we only calculated the standard deviation of the mechanical compression experiment, as shown in Table 3 (line 263).

(3) What is the cell behavior in the material suface for the new roughtness created after fabrication?

Response: Any processing method will produce a rough surface. The four sets of scaffolds we designed have also produced micron and nano-scale roughness on the surface after preparation, as shown in the figure below. The study found that the rough surface has a certain influence on the behavior of the cells [Small 10 (16) (2020) 1905422], but our main work in this article is to evaluate the macro-mechanical properties and permeability of the Ti6Al4V scaffold, without considering the cells on the rough surface. Thank you for your suggestion. This is our next focus of research.

(4) Do you know the integration of this porous materials and bone. Have some background?

Response: We have a certain degree of cooperation with many hospitals, so we have a certain understanding of the combination of porous materials and bones. Our goal is to determine the specific location of the bone defect, the size of the defect and the shape of the defect through CT scans, design the scaffold to the corresponding size, and then fix it by clinical methods such as steel plate.

(5) Why do you use Von misess stress and not the major principal stress.

Response: Von mises is the fourth strength theory. According to the energy conservation theory, it is generally used to judge the yield behavior of materials with better ductility. The Ti6Al4V material used in this article has better ductility. So, we think it is more reasonable to adopt Von misses. In addition, we refer to the research methods of some other researchers, and basically use Von mises. [Materials Science and Engineering C 43 (2014) 587 – 597; Materials Science and Engineering A 745 (2019) 195 – 202; Composites Part B 162 (2019) 154 – 161; et al] 

(6) How is the differences found with bone tissue in term of permeability and mechanical properties. Do you know the zone in the body where do you can implant the scafolds?

Response: Through CT tomography, combined with the processing of tomographic data by Mimics software, you can clearly observe the microstructure differences between the bone tissue layers. In the same fragment layer, the density of each part of the tissue, the size and shape of the pores are different. Or by preparing tissue sections and coating them with metal films, under an electron microscope, the differences between the tissues can be clearly observed. These differences represent differences in mechanical properties and permeability. Then, many researchers have determined the yield strength, elastic modulus, and permeability range of bone tissue through experiments. This provides us with a data reference for preparing the bionic bone scaffold structure. Strictly speaking, we believe that this kind of bionic scaffold can be implanted in any defect position of the human lower limb femur and tibia, which is also the significance of our research. But in order to achieve better results, we may fine-tune the scaffold parameters according to the specific location. For example, a doctor suggested to us that we hope to replace the femoral head necrosis area with our scaffold structure. We only need to do some specific analysis and adjustments based on the existing structure.

(7) Another minor recommendations are: In the introduction and discussion I recommend introduce more studies based on tissue engineering. Similar to theses examples:https://www.sciencedirect.com/science/article/abs/pii/S0142961209007777

https://pubmed.ncbi.nlm.nih.gov/16003400/.

Response: Thanks for your comments, we have added more studies based on tissue engineering in the introduction and discussion. (Page 3, Lines 46-50; Page 21, Lines 330-373)

(8) The color scale values must be corrected; removing (e + 000) and correcting the font size.

Response: We have revised it according to your comments. (Page 19, Fig. 9)

Reviewer #2: In this paper, the authors proposed a porous scaffold with elliptical pore shape, which is considered as similar to cancellous bone in terms of permeability and mechanical strength. A combination of computational and experimental approaches have been used by the authors for characterising the permeability and mechanical compressive strength. This is a thorough and well-designed study. The outcome from this study could be useful for the R&D of bone implants. However, the following concerns / comments need to be properly addressed before accepting for publication.

(1) Line 10 – 11: please specify compact bone or cancellous bone.

Response: Thanks for your comments, we have revised it according to your comments. (Page 1, Line 10)

(2) Line 36 – 45: I would suggest to summarise those studies in a table.

Response: The papers we read and the research results of other researchers are basically summarized and described in specific paragraphs, so that the expression is more convenient and flexible. We hope to retain this expression format, because it does not affect the specific content. We hope to get your approval.

(3) Line 47 – 49: What is the specific limitations of these previous studies? Why the bone cell behaviours will be different within the scaffold structures that are different from native bone?

Response: The specific limitations, we believe that these scaffold structures are certainly not optimal in terms of materials and structures. On the existing basis, their research may be considered very prominent, but after all, there are still differences between the structure and the real bone tissue of the human body. In the case of cube structure or diamond cubic structure, the structure is different from the cancellous bone structure, so the living space of cells is also different. There will be differences in nutrient delivery and waste excretion, which will affect the subsequent tissue regeneration effect. We can't say that our design is optimal, we can only say that we have made new explorations in the design of bone scaffold structure. In the future, we may design several other structures. We will use the known best-performing structures in all designs for animal experiments and further clinical experiments If a good repair effect can be achieved, then we can say that the product we designed is more mature.

(4) In the introduction section, I don't think Fig. 1 is necessary, as it is commonly known in the biomedical field.

Response: We agree with you. The reason we add this picture is to worry that the reviewers are not experts in the biomedical field, so that they can understand. We have removed Fig. 1 according to your comments. 

(5) Line 58 – 59: Please justify why you chose to investigate these 2 parameters?

Response: As a substitute for native bone tissue, the bionic bone scaffold should have the performance characteristics of native bone tissue. Among the two most important and basic performance characteristics, one is load-bearing performance and the other is material transmission performance. The load-bearing performance can be evaluated by mechanical tests, and the material transmission performance can be measured by permeability and pressure drop [Journal of the Mechanical Behavior of Biomedical Materials 93 (2019) 158-169]. So, these two parameters were selected.

(6) Line 65: “static compression experiment” is repetitive.

Response: Thanks for your comments. It was our negligence, forgot to add a statement. Of course, the static compression test is repeated. Three samples were selected for each group of experiments, and repeated experiments were carried out under the same conditions. We have revised it as required. (Page 9, Line 154)

(7) In section 2.3 – Simulation analysis of mechanical compression test: the mesh information of the FE model is missing.

Response: Thanks for your comments. We ignored the grid information for the sake of brevity, and now we have added it as required. (Page 6, Lines 118-120; Page 7, Fig. 2)

(8) In section 2.7 – Fluid dynamic simulation analysis: the flow properties, mesh and solver information are missing in this section, for example,

1) Flow property: is the flow laminar or turbulent? What is the Re number?

Response: The flow is laminar. R_e=(υ∙ρ∙d)/μ. Since the entrance velocity is extremely small, the calculated value of Re is very small, so in general, the default is laminar flow. Of course, we have revised the text as required. (Page 10, Lines 165-167)

2) Mesh: what is the mesh strategy that is used for this CFD model? Why did you choose that mesh strategy? Have you done any mesh sensitivity analysis? If yes, what are the results?

Response: Because the scaffold does not allow the use of hexahedrons for effective meshing, we use linear tetrahedrons for meshing. We did not do a mesh sensitivity analysis, but only did a few pre-analysis. When the size of our mesh division is relatively large, the result cannot be converged. We set the maximum mesh size of the scaffold to 0.001mm, and the number of mesh divided by each group of scaffolds is as high as several million. This is almost the most accurate result that our computer can handle. After several hours of calculation, convergence can be achieved. (Page 10, Lines 176-177)

3) Solver: which numerical method is used, finite element method / finite volume method / finite difference method? What is convergence criteria of the CFD model?

Without clarifying these, it is hard to assess the accuracy of your CFD results.

Response: We used finite volume method. The residual sensitivity criterion for convergence was set as 1e–5. (Page 10, Lines 167-168)

(9) In equation (4), please distinguish the vector, tensor, scalar in the equation. In Navier-Stokes equation, the expression of divergence of velocity seems wrong.

Response: Thanks for your comments, we have revised it according to your comments. (Page 10, Lines 170-171)

(10) In Fig.6 – BCs of CFD analysis: why would you define the inlet with L/2 away from the Plane A, and outlet at the Plane B?

Response: Adding the area above the plane A only considering the influence of the boundary effect, and refers to the setting methods of some documents [Acta Biomaterialia 112 (2020) 298-315; Materials and Design 192 (2020) 108754], which has been explained in line 175 of the article. There is no regulation for the size L/2, and there are many that ignore boundary effects and directly define the inlet as plane A. 

(11) In Fig. 9 and 10: in mechanical compression (both numerical and experimental), please justify why you chose the strain range of 0% - 9%? Are you sure bone can withstand such high strain, if you want to mimic the physiological level?

Response: The strain range of 0% - 9% in the experiment is not set by us, we just defined the compression speed as 0.05mm/min, and finally through the data processing, the stress-strain curve and the amount of deformation are obtained. The strain range in the simulation process is to make it similar to the compression test, so that the results of the experiment and the simulation can be better compared. Due to the excellent ductility of the titanium material sample, the strain range of the sample is very high during a complete compression process during mechanical compression [Acta Materialia 158 (2018) 354-368]. We are not to determine whether the bone can withstand such a large pressure and produce such a large deformation, but to judge how much pressure and deformation range our sample model can withstand. 

(12) I suggest to combine Fig. 10 and fig. 9 together into one figure, which will more clearly show the dis/agreement between numerical and experimental results.

Response: Thanks for your comments, we have made correction according to your comments. (Page 16, Fig. 8)

(13) Line 295 “transmission performance”: Do you mean nutrient diffusion? In nutrient diffusion, permeability will influence the diffusivity. But permeability itself can not be used for directly characterising the diffusion. Please revise this sentence.

Response: It is not accurate enough of our expression. We want to express that Permeability quantifies the ability of a porous medium to transmit fluid through its interconnected pores or channels when subjected to pressure. Now, we have revised it as required. (Page 18, Lines 300-301)

(14) Line 299 – 301: should this be in discussion section?

Response: Thanks for your comments, we have made correction according to your requirement. (Page 22, Lines 362-363)

(15) Line 304 – 307: How can you reach this point of migration of nutrient? Which parameter is it based on? If you are based on fluid velocity, I only can see that the local fluid velocity is changed with the change of local pore geometric features (pore size, porosity, pore shape), but can not see how this can influence the overall nutrient delivery throughout the scaffold.

Response: What we want to express is that the fluid flow rate gradually increases from the inlet to the outlet, which may accelerate the transportation of nutrients and the removal of waste. This is clearly pointed out in section 3.2.2 of " Acta Biomaterialia 112 (2020) 298-315" and section 3.3 of "Journal of the Mechanical Behavior of Biomedical Materials 93 (2019) 158-169". Maybe our expression was not accurate enough and we have made revisions, hoping to get your approval. (Page 18, Lines 308-311)

(16) After the results section, I would suggest to add a section of Discussion, in which the obtained results need to be compared to the references. Otherwise the current version of conclusion is not convincing.

Response: We have revised it according to your comments. (Page 21, Lines 330-373)

(17) Line 336: What does “beneficial to the migration and adsorption of nutrients to the scaffold” mean? Do you mean enhanced nutrient diffusion? Please clearly specify it.

Response: What we want to express is that the fluid flow rate gradually increases from the inlet to the outlet, which may accelerate the transportation of nutrients and the removal of waste. Please refer to the answer to Question 16. (Page 24, Line 386)

(18) Line 337 – 340: Without comparing with other already available scaffold geometries, how can you get this conclusion?

Response: There is no comparison with other already available scaffold geometries, but we have compared the yield strength, elastic modulus, permeability and other parameters of human cancellous bone. The scaffold we designed can match these parameters. Therefore, we believe that the scaffold we designed has the potential for further research and application. Of course, we may need to add a clearer discussion on this issue, which has been added in the article. (Page 21, Lines 330-373)

In addition, the funding information of this study, we have added it in the funding statement section.

We tried our best to improve the manuscript and made some changes in the manuscript. We appreciate for Editors/Reviewers’ warm work earnestly, and hope that the correction will meet with approval. Once again, thank you very much for your comments and suggestions.

Thank you and best regards.

Yours sincerely,

Chenglong Shi

Email: qhislcas@126.com

---

## [Decision Letter · Decision Letter 1]

18 Jan 2021

PONE-D-20-36743R1

Study on Mechanical Properties and Permeability of Elliptical Porous Scaffold Based on the SLM Manufactured Medical Ti6Al4V

PLOS ONE

Dear Dr. Shi,

Thank you for submitting your manuscript to PLOS ONE. After careful consideration, we feel that it has merit but does not fully meet PLOS ONE’s publication criteria as it currently stands. Therefore, we invite you to submit a revised version of the manuscript that addresses the points raised during the review process.

We look forward to receiving your revised manuscript.

Kind regards,

Jose Manuel Garcia Aznar

Academic Editor

PLOS ONE

Reviewers' comments:

Reviewer's Responses to Questions

**Comments to the Author**

1. If the authors have adequately addressed your comments raised in a previous round of review and you feel that this manuscript is now acceptable for publication, you may indicate that here to bypass the “Comments to the Author” section, enter your conflict of interest statement in the “Confidential to Editor” section, and submit your "Accept" recommendation.

Reviewer #1: All comments have been addressed

Reviewer #2: (No Response)

2. Is the manuscript technically sound, and do the data support the conclusions?

Reviewer #1: Yes

Reviewer #2: Yes

3. Has the statistical analysis been performed appropriately and rigorously? 

Reviewer #1: Yes

Reviewer #2: N/A

4. Have the authors made all data underlying the findings in their manuscript fully available?

Reviewer #1: Yes

Reviewer #2: No

5. Is the manuscript presented in an intelligible fashion and written in standard English?

Reviewer #1: Yes

Reviewer #2: Yes

6. Review Comments to the Author

Reviewer #1: (No Response)

Reviewer #2: Thanks for the author's effort in addressing the comments. Most of the comments have been properly addressed, but still have 2 minor ones left:

1. In the response to original comment (7), you gave the maximum mesh element size. So, this means that the element size is not uniform for the model geometry, right? If so, please could you specify which factors / parameters control the actual element size? This would be useful for the other researchers if they want to replicate the model.

2. In the response to original comment (9), although it was claimed that revision was made, it seems like no changes in the manuscript (e.g. the 2nd term of NS equation is still incorrect; units of parameters in NS equation are wrong). I know this will not influence the final results, as you use the commercial CFD software. However, once it is written in the paper, you have to make sure it is correct. So, please be careful when do the revision.

I would suggest to accept for publication after the 2nd revision is properly made.

7. PLOS authors have the option to publish the peer review history of their article (what does this mean?). If published, this will include your full peer review and any attached files.

Reviewer #1: **Yes: **Andy L. Olivares

Reviewer #2: No

---

## [Author Response · Author response to Decision Letter 1]

18 Jan 2021

Dear Editors and Reviewers:

Thank you for the reviewers’ comments concerning our revision entitled “Study on Mechanical Properties and Permeability of Elliptical Porous Scaffold Based on the SLM Manufactured Medical Ti6Al4V” (ID: PONE-D-20-36743R1). Those comments are all valuable and very helpful for revising and improving our paper, as well as the important guiding significance to our researches. We have studied the comments carefully and have made correction which we hope meet with approval. Revised portion are marked in blue in the paper. The main corrections in the paper and the responds to the reviewer’s comments are as flowing: 

Responds to the reviewer’s comments:

Reviewer #1: (No Response)

Reviewer #2: Thanks for the author's effort in addressing the comments. Most of the comments have been properly addressed, but still have 2 minor ones left:

I would suggest to accept for publication after the 2nd revision is properly made.

(1) In the response to original comment (7), you gave the maximum mesh element size. So, this means that the element size is not uniform for the model geometry, right? If so, please could you specify which factors / parameters control the actual element size? This would be useful for the other researchers if they want to replicate the model.

Response: Yes, the element size is not uniform for the model geometry. Because the geometry is not the same at every position, some parts are weaker, and some edge parts are finely divided. We defined the approximate global seed size as 0.03mm and the maximum size as 0.05mm. We understand that the meshing you mentioned will have an impact on the results. Since we conducted a mechanical compression test, the simulation results are more of an approximate reference. Of course, we have made amendments as required. (Page 6, Lines 119-121)

(2) In the response to original comment (9), although it was claimed that revision was made, it seems like no changes in the manuscript (e.g. the 2nd term of NS equation is still incorrect; units of parameters in NS equation are wrong). I know this will not influence the final results, as you use the commercial CFD software. However, once it is written in the paper, you have to make sure it is correct. So, please be careful when do the revision.

Response: Thanks for your comments, we have revised it according to your comments. (Page 10, Lines 171-173)

We tried our best to improve the manuscript and made some changes in the manuscript. We appreciate for Editors/Reviewers’ warm work earnestly, and hope that the correction will meet with approval. Once again, thank you very much for your comments and suggestions.

Thank you and best regards.

Yours sincerely,

Chenglong Shi

Email: qhislcas@126.com

---

## [Decision Letter · Decision Letter 2]

28 Jan 2021

PONE-D-20-36743R2

Study on Mechanical Properties and Permeability of Elliptical Porous Scaffold Based on the SLM Manufactured Medical Ti6Al4V

PLOS ONE

Dear Dr. Shi,

Thank you for submitting your manuscript to PLOS ONE. After careful consideration, we feel that it has merit but does not fully meet PLOS ONE’s publication criteria as it currently stands. Therefore, we invite you to submit a revised version of the manuscript that addresses the points raised during the review process.

Please, revise eq 4 according to comments from reviewer 2.

We look forward to receiving your revised manuscript.

Kind regards,

Jose Manuel Garcia Aznar

Academic Editor

PLOS ONE

Reviewers' comments:

Reviewer's Responses to Questions

**Comments to the Author**

1. If the authors have adequately addressed your comments raised in a previous round of review and you feel that this manuscript is now acceptable for publication, you may indicate that here to bypass the “Comments to the Author” section, enter your conflict of interest statement in the “Confidential to Editor” section, and submit your "Accept" recommendation.

Reviewer #1: All comments have been addressed

Reviewer #2: (No Response)

2. Is the manuscript technically sound, and do the data support the conclusions?

Reviewer #1: Yes

Reviewer #2: Yes

3. Has the statistical analysis been performed appropriately and rigorously? 

Reviewer #1: Yes

Reviewer #2: N/A

4. Have the authors made all data underlying the findings in their manuscript fully available?

Reviewer #1: Yes

Reviewer #2: No

5. Is the manuscript presented in an intelligible fashion and written in standard English?

Reviewer #1: Yes

Reviewer #2: Yes

6. Review Comments to the Author

Reviewer #1: (No Response)

Reviewer #2: Thanks for your time & effort on addressing my comments. My previous comment 1 has been addressed and changes were made in manuscript. That's good. However, the comment 2 was not addressed thoroughly. You did not revise eq. 4, which I have already mentioned twice in previous comments. According to that stated in paper, you want to express the incompressible flow. From continuity equation, the incompressible flow has the divergence of velocity of 0 (∇ * u= 0). However, what you wrote is an incomplete equation, advection of something unknown = 0. I don't know which scalar / vector you want to put in. But please use divergence of velocity = 0 (i.e. ∇ * u= 0) for incompressible flow in Eq. 4, if that is what you want to express. I know this minor problem will not influence the results, but it is better to be corrected before publishing.

7. PLOS authors have the option to publish the peer review history of their article (what does this mean?). If published, this will include your full peer review and any attached files.

Reviewer #1: **Yes: **Andy L. Olivares

Reviewer #2: No

---

## [Author Response · Author response to Decision Letter 2]

29 Jan 2021

Dear Editors and Reviewers:

Thank you for the reviewers’ comments concerning our revision entitled “Study on Mechanical Properties and Permeability of Elliptical Porous Scaffold Based on the SLM Manufactured Medical Ti6Al4V” (ID: PONE-D-20-36743R2). Those comments are all valuable and very helpful for revising and improving our paper, as well as the important guiding significance to our researches. We have studied the comments carefully and have made correction which we hope meet with approval. Revised portion are marked in blue in the paper. The main corrections in the paper and the responds to the reviewer’s comments are as flowing: 

Responds to the reviewer’s comments:

Reviewer #1: (No Response)

Reviewer #2: Thanks for your time & effort on addressing my comments. My previous comment 1 has been addressed and changes were made in manuscript. That's good. However, the comment 2 was not addressed thoroughly. You did not revise eq. 4, which I have already mentioned twice in previous comments. According to that stated in paper, you want to express the incompressible flow. From continuity equation, the incompressible flow has the divergence of velocity of 0 (∇ * u= 0). However, what you wrote is an incomplete equation, advection of something unknown = 0. I don't know which scalar / vector you want to put in. But please use divergence of velocity = 0 (i.e. ∇ * u= 0) for incompressible flow in Eq. 4, if that is what you want to express. I know this minor problem will not influence the results, but it is better to be corrected before publishing.

Response: Thanks for your comments, we have studied the comments carefully and have made correction which we hope meet with approval. (Page 10, Lines 171-172)

We tried our best to improve the manuscript and made some changes in the manuscript. We appreciate for Editors/Reviewers’ warm work earnestly, and hope that the correction will meet with approval. Once again, thank you very much for your comments and suggestions.

Thank you and best regards.

Yours sincerely,

Chenglong Shi

Email: qhislcas@126.com

---

## [Decision Letter · Decision Letter 3]

15 Feb 2021

Study on Mechanical Properties and Permeability of Elliptical Porous Scaffold Based on the SLM Manufactured Medical Ti6Al4V

PONE-D-20-36743R3

Dear Dr. Shi,

We’re pleased to inform you that your manuscript has been judged scientifically suitable for publication and will be formally accepted for publication once it meets all outstanding technical requirements.

Kind regards,

Jose Manuel Garcia Aznar

Academic Editor

PLOS ONE

Additional Editor Comments (optional):

Reviewers' comments:

Reviewer's Responses to Questions

**Comments to the Author**

1. If the authors have adequately addressed your comments raised in a previous round of review and you feel that this manuscript is now acceptable for publication, you may indicate that here to bypass the “Comments to the Author” section, enter your conflict of interest statement in the “Confidential to Editor” section, and submit your "Accept" recommendation.

Reviewer #2: All comments have been addressed

2. Is the manuscript technically sound, and do the data support the conclusions?

Reviewer #2: Yes

3. Has the statistical analysis been performed appropriately and rigorously? 

Reviewer #2: (No Response)

4. Have the authors made all data underlying the findings in their manuscript fully available?

Reviewer #2: No

5. Is the manuscript presented in an intelligible fashion and written in standard English?

Reviewer #2: Yes

6. Review Comments to the Author

Reviewer #2: The authors have addressed all of my comments. I have no further comments. Therefore, I would recommend to accept it for publication.

7. PLOS authors have the option to publish the peer review history of their article (what does this mean?). If published, this will include your full peer review and any attached files.

Reviewer #2: No

---

## [Editor Report · Acceptance letter]

22 Feb 2021

PONE-D-20-36743R3 

Study on Mechanical Properties and Permeability of Elliptical Porous Scaffold Based on the SLM Manufactured Medical Ti6Al4V 

Dear Dr. Shi:

I'm pleased to inform you that your manuscript has been deemed suitable for publication in PLOS ONE. Congratulations! Your manuscript is now with our production department. 

Kind regards, 

on behalf of

Dr. Jose Manuel Garcia Aznar 

Academic Editor

PLOS ONE